# A comprehensive analysis of COVID-19 vaccination behavior: The influence of religion, information sources, political leanings, and demographic factors

Defne Över 🔵*, Emilce Santana, Ernesto F. L. Amaral 🔵, Chaitanya Lakkimsetti, Anna Estelle Kelley, Dulce Angelica Espinoza

Department of Sociology, Texas A&M University, College Station, Texas, United States of America

* dover@tamu.edu

## Abstract

The COVID-19 pandemic highlighted the crucial role of vaccines in controlling the virus. Despite their effectiveness, however, vaccine hesitancy remained a challenge, particularly within certain population groups. This multi-disciplinary study investigates the diverse socio-demographic factors influencing COVID-19 vaccination decisions in the United States. Through a nationally representative survey of 5,240 people, the research explores the interplay of information sources, religious beliefs, political party, and demographic characteristics of the respondents. Our findings reveal associations of main sources of information with vaccination likelihood, with the Centers for Disease Control and Prevention demonstrating the highest association with full vaccination. Religious beliefs are significant determinants, with Evangelical Protestants exhibiting the lowest vaccination rates. We also highlight the intricate relationship between political leanings and vaccination behavior, emphasizing higher levels of vaccination among Democrats. Demographic variables, including age, education, gender, and race/ethnicity, also play pivotal roles, exposing disparities in vaccination access and decisions. In particular, older individuals and those with higher levels of education show a greater inclination to achieve full vaccination, while women and African Americans are less likely to attain complete vaccination. Lastly, while major ethnoracial groups seem to respond to different sources of information similarly, there are also nuanced differences, such as Asians being especially likely to be fully vaccinated if they depend on the CDC or other health sources while more disadvantaged groups seem less responsive to these sources. Overall, this research provides a comprehensive analysis of the nuanced factors shaping vaccination behavior. It contributes valuable knowledge to public health strategies, emphasizing the need for targeted communication campaigns tailored to diverse communities.

**Data availability statement:** All relevant data are within the manuscript and its Supporting Information files.

**Funding:** This research was supported by two seed grant programs from Texas A&M University: 243636-00000 Innovation X Grant Program (Co-PIs: Amaral, Over, Santana, Lakkimsetti); T3 Program (Co-PIs: Over, Lakkimsetti). These programs were discontinued and do not have URLs. The funders did not play any role in the study design, data collection and analysis, decision to publish, or preparation of the manuscript.

**Competing interests:** The authors have declared that no competing interests exist.

## Introduction

During the COVID-19 pandemic, vaccines emerged as a highly effective method of providing protection against the wild type SARS-CoV-2 [1,2]. Medical reports published during the pandemic established the role of vaccines in limiting the spread of the virus [3]. COVID-19 vaccines thus provided a major opportunity to return to normalcy if a significant number of people got vaccinated. Yet, vaccination refusal and hesitancy within communities remained a major impediment on the road to normalcy [4]. In many places in the United States where COVID-19 vaccines were widely accessible, vaccination rates remained below the desired threshold, even when infection rates in the given community were high [5–7]. Focusing on the case of the United States, this mixed method study examines key factors influencing vaccination decision-making during the COVID-19 pandemic.

Individuals adopt different behaviors toward vaccination. Some are complete acceptors, others total refusers and there are also hesitant individuals who may refuse some vaccines, but agree to others; delay vaccines; or accept vaccines but are unsure of doing so [8]. Researchers recognize that factors influencing vaccine behavior vary depending on time, place, and the type of vaccine [9–11], and trace the variation in vaccine acceptance to reasons such as lack of trust in vaccine administrators; individual and group level perceptions of the vaccine; and the characteristics of the vaccine [12,13].

Research on other vaccination programs, e.g., World Health Organization (WHO)'s Expanded Program on Immunization in India, the state-mandated childhood vaccines in the US, and human papillomavirus (HPV) vaccines in the US, for instance, has shown that viewing vaccination as part of a political agenda, perceiving health as a matter of individual choice rather than public decision making, personalization of risks associated with vaccination, presence of misinformation and organized vaccine resistance in one's community increased the likelihood of vaccine hesitancy and refusal in a given community [9,14–18]. Research on the swine influenza vaccine in the UK has shown that resourceless and marginalized peoples and disadvantaged minority groups are less likely to get vaccinated [19].

While situations under which we observed COVID-19 vaccine hesitancy vastly differ from vaccine hesitancy for children's diseases, HPV, or influenza [20], there are parallels in ambiguities experienced in relation to vaccine safety, and lack of trust in the state, and public health institutions in certain communities, showing that vaccination is not only an individual but also a community decision. Recognizing the impact of community membership on individuals' vaccination decisions, this article examines how people with different socioeconomic, demographic, political, and religious backgrounds responded to COVID-19 vaccination campaigns in the US. Specifically, we inquire into the connections between beliefs (religious, and political), sources of information (religious leaders, health professionals, the media, political actors), community characteristics (age, level of education, race/ethnicity, gender), and individuals' vaccine behaviors in the COVID-19 pandemic.

The study is based on a nationally representative survey with national quotas for age, sex, race/ethnicity, and political party affiliation, distributed to 5,240 adults in the

United States. In what follows, we present the scholarly literature on vaccination behavior in the US during the COVID-19 pandemic, our data collection and analysis methods, and the results of our analysis. We conclude the article with a discussion on the implications of our findings for future vaccination campaigns, in particular for communication strategies aimed at increasing vaccination uptake.

## Vaccine behavior in the US during the COVID-19 pandemic

The COVID-19 pandemic has affected communities across the US disproportionately. In Texas, for example, with 40% Hispanic and 40% White population, the death toll in the Hispanic community was three times that of the White population when adjusted for age [21]. Despite the higher risks encountered by minority groups, research has found that there exist racial and ethnic differences in vaccine hesitancy and refusal [22–25], with minoritized groups reporting higher levels of vaccine hesitancy [26–32]. African Americans and Hispanics, in particular, have led racial and ethnic groups in vaccine hesitancy while Whites had the highest share of eligible people who received vaccination. For example, while the His- panic population had higher chances of morbidity, hospitalization, and death due to COVID-19, only 25% of Hispanics had received a vaccine shot in Texas compared to 50% of Whites by March 15, 2021 [5]. In the same period, over 30% of Hispanic adults wanted to "wait and see" how the vaccine works for other people, and 20% refused to vaccinate unless required to do so [6]. Some researchers posited institutional trust as a reason for lower levels of vaccination among minority groups–particularly African Americans [23,25,29,33,34]. Accordingly, African Americans were more likely to refuse vaccination than Whites for the sole reason that they do not trust COVID vaccine to be safe or effective [31,33,35]. This trend in vaccine attitudes remained despite the fact that African Americans have widely been found more likely to be at risk of and express concern for the virus than Whites [26,29].

Studies looking at the relationship between sex and gender and vaccine uptake have, on the other hand, shown mixed results. Research conducted early in the pandemic showed that women expressed greater vaccine hesitancy than men [27,36,37]. A scoping review of the literature on vaccine uptake in the first two years of the pandemic showed that while men did tend to have greater vaccine uptake intentions, there were limited differences in actual uptake [38]. Studies conducted in later years of the pandemic continued to show greater vaccine hesitancy among women than men, but data collected at vaccination sites indicate that women have higher vaccination rates than men [25].

In terms of age, older adults were more likely than younger adults to express willingness to be vaccinated, and unlike gender, this was reflected in vaccine uptake [25,27,28,39,40]. Young adults were also more likely to report being unwill- ing or undecided about getting the vaccine [27]. A survey conducted at a mid-sized public university in Connecticut, for example, revealed that about 30% of college students indicated vaccine refusal, and about 20% of college students were unsure about whether or not to get vaccinated once they were eligible [7]. At the same time, those with higher levels of education were more likely to report higher levels of COVID-19 vaccine acceptance. Early in the pandemic, those with college or graduate degrees were more likely to report willingness to receive the vaccine than those with lower levels of educational attainment [27,30,37]. Those with higher levels of education were also more likely to have actually gotten the vaccine once it became available [39]. Additionally, those with higher income levels had higher levels of vaccine accep- tance and trust, while those with lower income levels were more likely to report vaccine hesitancy [25,28].

To explain the variation in vaccine uptake across groups, some scholars approached vaccination behavior as a political decision. Exploring the effect of partisanship, conservatism, and Trumpism as factors driving vaccine refusal or hesitancy, they found that political conservatism is significantly associated with greater skepticism toward COVID-19 in general [41,42]. Researchers further indicated that being more politically conservative is a consistently negative predictor of vac- cination likelihood [25,43]. While some [43] have argued that the partisan divide in vaccine hesitancy has increased over time, others [25] have shown that political conservatism became a consistent significant negative predictor of trust in the vaccine starting in November 2020, when the U.S. elected Joe Biden as President. Others confirmed these findings by showing that democrats are more likely to have gotten vaccinated against COVID-19 than other adults [39]; that increased

Republican affiliation correlated with lower vaccination rates [44]; and that there is a negative relationship between Trump support and COVID-19 vaccination rate [45,46].

Further research into the relationship between political identity and vaccine uptake explored how political identity interacted with other factors in predicting attitudes towards the COVID-19 vaccine. Researchers have found that support for Trump and conservatism were strong predictors of negative attitudes toward vaccination [47]. They have also found that type of religious identification, in particular identification with Christian nationalism, was significant for vaccine hesitancy [42]. These researchers also suggested that Christian nationalism was a strong predictor that individuals would be less likely to engage in precautions such as avoiding crowded venues, wearing a mask, and other protective public health practices. Others have similarly proposed that Christian nationalism as a pervasive ideology that rejects scientific authority and promotes allegiance to conservative political leaders was one of the two strongest predictors of anti-vaccine attitudes, stronger than political or religious characteristics considered separately [48].

Digging deeper into the variation in vaccination behaviors of racial, ethnic, gender, age and political communities, scholars have pointed at information sources as a significant factor driving vaccination decisions [54]. Accordingly, social media platforms such as Facebook and Twitter contributed to the unregulated spread of misinformation about COVID-19 [49]. Interaction with negative tweets about COVID-19, in particular, was associated with expressions of lack of confidence in vaccine safety, distrust in governments and experts, and widespread misinformation or rumors [50]. Scholars also found that conservative media outlets, along with political and thought leaders, played a crucial role in downplaying the virus and discouraging adherence to recommended public health guidelines [45,49,51]. Followers of conservative news networks such as *Fox News*, those who cited Trump, his task force, and personal and community networks as their favored COVID-19 news sources were found to be far less likely than those who relied on other information sources to have received at least one shot of the vaccine. Similarly, those who reported seeking information from religious leaders were less likely to be vaccinated [52]. In contrast, those that got information from official government and health agencies, healthcare providers as well as those who followed more mainstream news sources were more likely to take the COVID-19 pandemic seriously [53]. For example, those that got COVID-19 related information from *The New York Times* and commercial news sources like *CNN* or *MSNBC* reported greater concern about contracting COVID-19 [54].

Building on existing research, the current study examines the impact of race and ethnicity, gender, age, political and religious beliefs, and information sources on COVID-19 vaccination decisions. It extends these studies by investigating how individuals' racial and ethnic backgrounds interact with their reliance on specific information sources to shape vaccination choices. Unlike prior research, which typically focuses on specific ethnic and racial groups' vaccination decisions, by relying on a nationally representative sample, this study explores whether different racial and ethnic groups rely on different information sources and whether this reliance influences their vaccination behavior in varying ways. The study thus aims to offer empirically grounded strategies for the use of information sources in the communication of public health messages to specific communities.

Our main exploratory hypotheses are the following:

Hypothesis 1: More vulnerable populations (e.g., ethno-racial minorities, women, elderly, lower-educated individuals) face obstacles to follow public health practices due to several factors (e.g., being overrepresented in essential jobs, having less access to government benefits) and are less likely to be vaccinated against COVID-19.

Hypothesis 2: Individuals who have conservative political and religious beliefs are less likely to follow these practices and to be vaccinated against COVID-19, compared to individuals who have liberal political and religious beliefs.

Hypothesis 3: Individuals who receive COVID-19 related information primarily from their informal networks (family, employer, friends) are less likely to be vaccinated against COVID-19, compared to individuals who primarily follow health experts as information sources.

                                                                           

## Methods

This research project is registered at Texas A&M University Institutional Review Board with the IRB ID: IRB2020-0722D. We started this project with a qualitative pilot study undertaken in Brazos County, Texas in 2021. For the pilot study, we conducted interviews via phone with 44 adults aged 18 and above. The research team obtained verbal consent from the interviewees using an IRB-approved consent script. During each interview session, participants were prompted to answer 31 open-ended questions about their racial, religious, political, socio-economic characteristics, their lived experiences of the pandemic, and their strategies of coping with COVID-19 including social distancing and mask wearing. The interviews highlighted the variation in the level of peer pressure, lived experiences of COVID-19 pandemic, and belief in science across age and gender groups, ethnic and racial communities, and socio-economic status. The interviews pointed at a major variation in the sources that people used to gather information on COVID-19. While 17.7% of interviewees mentioned using news apps as main sources of information related to the COVID-19 pandemic, 24.4% claimed to have gathered information primarily from their friends and family, and 57.7% used social media as a news source of information. The interviews also revealed that strategies of coping with the conditions created by COVID-19 along with the motivations behind these strategies varied significantly among the interviewees.

Based on the findings of the qualitative pilot study, we designed a nationally representative survey with questions about factors driving individuals' COVID-19 vaccine decisions in the United States. At the beginning of the survey, respondents read out the informed consent document. If they agreed with the document, the online survey was administered. The survey was fielded solely online and was distributed between September 5, 2023, and September 9, 2023, via the Prime Panel service from CloudResearch. Prime Panels are an "aggregation of opt-in market research panels" [55]. More specifically, multiple suppliers bid to CloudResearch for the opportunity to supply respondents to the specified samples requested by different research teams [56]. The benefit of this approach is that our research team was able to sample subpopulations that may be difficult to reach [55,56]. CloudResearch Prime Panels are an advancement upon other data collection softwares, such as Amazon Mechanical Turk, because it is more demographically targeted and thus able to be more representative. Participants are compensated in a number of ways, based on the platform through which the individual participates. It includes compensation methods such as reward points, gift cards, or cash. Our team was able to assign sampling quotas to ensure a representative sample [55,56]. Our team selected respondents according to national quotas available in the CloudResearch Prime Panel tool, according to the US population distributed by age, sex, race, ethnicity, and political party. Respondents were provided by the suppliers who submitted their bids to this research project with Cloud Research. The quota for age considered the following groups: 18–29 (21.62% of respondents), 30–44 (26.55%), 45–59 (24.76%), and 60–99 (27.07%). The quota for sex, considered female (52.00%), and male (48.00%). For race, the quota selected respondents to reach this distribution: White (75.60%), African American (11.80%), Asian (4.73%), American Indian/Alaska Native (1.79%), and other races (6.10%). The quota for ethnicity utilized these two groups: Hispanic (17.44%) and not Hispanic (82.58%). For political party affiliation, the quota selected respondents according to three groups: Democrat (38.85%), Republican (34.31%), and Independent (26.84%). CloudResearch received a total of 5,246 respondents distributed in the quotas listed above. These respondents answered 100% of the questions and there were no duplicates. This data collection strategy does not provide a response rate as in traditional data collection methods. For this specific paper, we kept responses from respondents who identified as either male or female in the assigned sex at birth category, which resulted in dropping six responses, and a total of 5,240 respondents.

The survey captured respondents' vaccine behaviors alongside their community characteristics, the sources that they have received information from, their perceptions on a variety of issues ranging from politics to science and vaccination, and their lived experiences of COVID-19 pandemic. Our questionnaire had a total of 88 questions related to 12 blocks: (1) COVID-19 experience; (2) COVID-19 vaccine behavior; (3) views on science, vaccines, and the COVID-19 pandemic; (4) sources of information about the COVID-19 pandemic and vaccines; (5) misinformation and fake news about the COVID-19 pandemic; (6) gender and domestic work; (7) information about current residence; (8) demographic characteristics; (9)

religious beliefs; (10) political background; (11) main occupation; and (12) income. More specifically, we collected information on behavioral, socioeconomic, and demographic measures such as race/ethnicity, social class, gender, educational attainment, religion, occupation, health insurance, sick leave, political ideology, migration status, membership to community associations, and consumption of media sources.

## Statistical analysis

To test our hypotheses about likelihood of being vaccinated against COVID-19, we selected a variable about vaccination as the dependent variable for a logistic regression model. Out of 5,240 individuals in our sample, 3,644 were fully vaccinated, 389 were partially vaccinated, and 1,207 were unvaccinated. For this study, we combined individuals who were partially vaccinated and those who were not vaccinated for COVID-19 to compare with those who were fully vaccinated.

We also tested different specifications of the dependent variable. For instance, we estimated logistic regression models by combining fully and partially vaccinated individuals to compare with those not vaccinated. Additionally, we estimated a multinomial logistic regression that utilized the original variable with three categories related to COVID-19 vaccination by estimating in the same model comparisons between partially vaccinated and fully vaccinated, as well as between unvaccinated and fully vaccinated. These additional specifications were not different from the analysis provided in this study.

Our overall rationale is that at that time of our survey in September 2023, individuals who were not fully vaccinated made the decision to be either partially vaccinated or unvaccinated. Thus, we focus the analysis on logistic regressions that estimate the chances of individuals being fully vaccinated, controlling for a series of independent variables.

We test Hypothesis 1 about more vulnerable populations being less likely to be vaccinated against COVID-19 with a series of sociodemographic variables: Race/ethnicity, assigned sex at birth; age groups; and education levels. This way we could verify if ethno-racial minorities, women, older individuals, and individuals with lower education levels are less likely to be vaccinated. The race/ethnicity groups utilized in our analysis are: (1) Non-Hispanic White; (2) Hispanic; (3) Non-Hispanic African American; (4) Non-Hispanic Asian; (5) Other race/ethnicity or two or more races/ethnicities. For assigned sex at birth, a total of six individuals reported an undetermined sex. For the current analysis, we only include individuals who reported male or female as the assigned sex at birth. Our age groups (18–29, 30–39, 40–49, 50–64, 65–74, 75+) follow the standard utilized by the CDC when reporting COVID-19 indicators. We classified the sample in the following education groups: (1) Less than high school; (2) High school; (3) Some college; (4) Associate degree; (5) Bachelor's degree; and (6) Graduate degree.

Our Hypothesis 2 states that individuals with more conservative political and religious beliefs are less likely to be vaccinated against COVID-19. In order to test this hypothesis, we included disaggregated categories for political parties and religious beliefs to explore detailed associations of these variables with COVID-19 vaccination. Political parties include these categories: (1) Strong Democrat; (2) Democrat; (3) Independent; (4) Republican; (5) Strong Republican; and (6) other parties. Religious preference includes the following categories: (1) Evangelical Protestant; (2) Other Protestant; (3) Catholic; (4) Other; (5) Agnostic or Atheist; and (6) Nothing in particular.

As a strategy to test Hypothesis 3 that individuals who follow family, friends and employer as their primary source of information on COVID-19 are less likely to be vaccinated against COVID-19, we included the main source of information about COVID-19 vaccines as an independent variable. This question has the following categories: (1) Centers for Disease Control and Prevention (CDC); (2) Employer; (3) Family and friends; (4) Food and Drug Administration (FDA); (5) Health insurers; (6) Hospital system websites (e.g., Kaiser Permanente); (7) Local health officials; (8) News sources (e.g., television, internet, and radio); (9) Nurses; (10) Pharmacists; (11) Primary care providers; (12) Professional organizations; (13) Religious leaders; (14) State health departments; (15) Online publishers of medical information (e.g., WebMD or Mayo Clinic); (16) Social media (e.g., Facebook, Twitter, Instagram); (17) Union leaders; and (18) Other sources. We grouped these sources of information based on how related they are to each other, as well as based on the percentage of respondents who were fully vaccinated for COVID-19 in each category. Based on these criteria, we simplified this variable into

five categories: (1) CDC, which had 83% of respondents fully vaccinated; (2) Other health sources, which includes categories that had between 53% and 73% of respondents fully vaccinated: FDA, health insurers, hospital system websites, local health officials, nurses, pharmacists, primary care providers, state health departments, online medical publishers; (3) Employer (58% of respondents fully vaccinated) and family and friends (47%) became one category; (4) News sources (66%) and professional organizations (73%) were combined; (5) Other sources, which includes categories that had between 25% and 40% of respondents fully vaccinated: Religious leaders, social media, union leaders, and other sources.

We examined the percentage distribution of respondents by all variables included in our analysis, as well as cross-tabulations between all independent variables and the dependent variable: Whether individuals are fully vaccinated for COVID-19 (Table 1). Then, we estimated a series of bivariate logistic regression models to analyze associations of each independent variable with the dependent variable (Table 2). Additionally, we estimated a multivariate logistic regression including all independent variables predicting the dependent variable (Table 3).

Finally, we verified that the percentage of respondents fully vaccinated by the main information source and race/ethnicity vary considerably (Table 4). In order to statistically test these variations, we estimated models to evaluate interaction effects between race/ethnicity and information source for COVID-19 vaccination. Table 5 includes these interactions as a set of dummies. Table 6 illustrates results from separate models for each race/ethnicity, which is equivalent to interacting race/ethnicity with all independent variables. The sample sizes are small for some sub-groups of respondents by the main source of information about vaccines and race/ethnicity (e.g., Non-Hispanic Asian and other race/ethnicity). Due to this limitation, for models on Table 6, we focus the analysis on Non-Hispanic White, Hispanic, and Non-Hispanic African American.

We conducted a power analysis using G*Power software to determine the required sample size for a logistic regression analysis examining factors associated with being fully vaccinated for COVID-19. The dependent variable indicates vaccination status, with 69.54% fully vaccinated and 30.46% not vaccinated, corresponding to an odds ratio of 5.212062 (0.6954/0.3046 * 0.6954/0.3046) under the alternative hypothesis. We used a two-tailed Z-test with an alpha error probability of 0.05 and a desired power of 0.95. We incorporated a binomial distribution for an additional independent variable with parameter π equals to 0.1. We accounted for multicollinearity with a Pseudo $R^2$ of 0.2 for other predictors, using information from the multivariate model in Table 3. Applying the large sample approximation and the Demidenko (2007) test procedure without variance correction, the analysis determined a critical z-value of 1.9599640 and a required total sample size of 313 to achieve an actual power of 0.9502377 [57]. These findings ensure our study with a sample of 5,240 respondents is adequately powered to detect meaningful associations between our independent variables and vaccination status.

The following section presents the results of our empirical findings. We illustrate a series of univariate, bivariate, and multivariate statistical findings to test our hypotheses. (See Supporting Information files (S1 and S2 Databases; S3 and S4 Codes) for data files and code).

## Main results

Tables 1 and 2 illustrate bivariate analyses between COVID-19 vaccination (dependent variable) and the independent variables: race/ethnicity, sex, age groups, education groups, political party, religious groups, and main source of information about COVID-19 vaccine. Specifically, Table 1 illustrates frequency distributions for the variables featured in the main analyses. This table also contains cross tabulations between COVID-19 vaccination and each of the independent variables. Further, Table 2 shows seven unadjusted logistic regression models (one model for each independent variable).

In addition to the main independent variable, COVID-19 vaccination status, we collected several demographic characteristics of respondents, including race/ethnicity, sex, age, and educational attainment. First, at the time of data collection in 2023, 69.54% of respondents were fully vaccinated and 30.46% of respondents were unvaccinated or partially vaccinated (Table 1). Moreover, the majority of respondents were non-Hispanic White. Our sample was comparably

**Table 1. Percentage distribution of respondents by variables of interest and cross-tabulation with COVID-19 vaccination status, 2023.**

| Variables | Column % | Fully vaccinated (row %) | Not fully vaccinated (row %) | Sample size | Association with vaccination |
|---|---|---|---|---|---|
| **COVID-19 vaccination status** | —— | 69.54 | 30.46 | 5,240 | —— |
| **Sex** | | | | | Chi square |
| Male | 47.50 | 73.72 | 26.28 | 2,489 | (df = 1): |
| Female | 52.50 | 65.76 | 34.24 | 2,751 | 39.15*** |
| **Age group** | | | | | Chi square |
| 18–29 | 19.39 | 59.55 | 40.45 | 1,016 | (df = 5): |
| 30–39 | 19.27 | 65.45 | 34.55 | 1,010 | 177.66*** |
| 40–49 | 17.27 | 63.76 | 36.24 | 905 | |
| 50–64 | 22.50 | 73.20 | 26.80 | 1,179 | |
| 65–74 | 14.62 | 81.59 | 18.41 | 766 | |
| 75+ | 6.95 | 85.99 | 14.01 | 364 | |
| **Education group** | | | | | Chi square |
| Less than high school | 3.17 | 40.36 | 59.64 | 166 | (df = 5): |
| High school | 25.13 | 56.04 | 43.96 | 1,317 | 394.77*** |
| Some college | 21.09 | 64.80 | 35.20 | 1,105 | |
| Associate | 12.00 | 68.36 | 31.64 | 629 | |
| Bachelor | 23.51 | 80.93 | 19.07 | 1,232 | |
| Graduate | 15.10 | 87.99 | 12.01 | 791 | |
| **Political party** | | | | | Chi square |
| Strong Democrat | 25.31 | 86.65 | 13.35 | 1,326 | (df = 5): |
| Democrat | 18.97 | 76.06 | 23.94 | 994 | 343.03*** |
| Independent | 20.13 | 61.14 | 38.86 | 1,055 | |
| Republican | 16.95 | 59.01 | 40.99 | 888 | |
| Strong Republican | 17.54 | 58.54 | 41.46 | 919 | |
| Other party | 1.11 | 55.17 | 44.83 | 58 | |
| **Religious group** | | | | | Chi square |
| Evangelical Protestant | 14.58 | 62.83 | 37.17 | 764 | (df = 5): |
| Other Protestant | 15.59 | 71.11 | 28.89 | 817 | 167.32*** |
| Catholic | 24.92 | 76.80 | 23.20 | 1,306 | |
| Other | 5.34 | 85.36 | 14.64 | 280 | |
| Agnostic/Atheist | 9.71 | 78.59 | 21.41 | 509 | |
| Nothing in particular | 29.85 | 60.17 | 39.83 | 1,564 | |
| **Race/ethnicity** | | | | | Chi square |
| White | 68.07 | 70.31 | 29.69 | 3,567 | (df = 4): |
| Hispanic | 13.21 | 68.35 | 31.65 | 692 | 37.61*** |
| African American | 11.15 | 63.01 | 36.99 | 584 | |
| Asian | 3.53 | 85.41 | 14.59 | 185 | |
| Other race/ethnicity | 4.05 | 64.62 | 35.38 | 212 | |
| **Main source of information about vaccines** | | | | | Chi square |
| Centers for Disease Control and Prevention (CDC) | 30.84 | 83.35 | 16.65 | 1,616 | (df = 4): |
| Other health sources | 49.71 | 68.06 | 31.94 | 2,605 | 392.84*** |
| Employer, family, and friends | 9.26 | 49.28 | 50.72 | 485 | |
| News sources and professional organizations | 6.51 | 67.16 | 32.84 | 341 | |
| Other sources | 3.68 | 29.02 | 70.98 | 193 | |

*(Continued)*

**Table 1.** (Continued)

Note:

*Significant at p<0.1;

**Significant at p<0.05;

***Significant at p<0.01.

Source: 2023 Survey about Effective Communication Strategies During the COVID-19 Pandemic.

representative of males and females. For the age variable, there were relatively similar percentages of respondents falling into age groups between 18 and 64. In terms of educational achievement, most respondents reported either a high school diploma, some college, or a bachelor's degree. Respondents were also asked to report their political party, including the strength of their association with that party, and religious identification. The sample distribution shows a substantial range in political party affiliations and strength. For religious identification, most respondents identified as some form of Protestant or as "nothing in particular. Lastly, the majority of respondents reported either the Centers for Disease Control and Prevention (CDC) or other health sources (i.e., the FDA, as well as respondent's health insurer, hospital system, local health, nurses, pharmacists, primary care, state health, and online medical) as their main source of information about vaccines.

According to the distribution of respondents by COVID-19 vaccination status (Table 1) and the unadjusted models (Table 2), except for age group, fully vaccination patterns by sociodemographic characteristics follow Hypothesis 1: More vulnerable populations are less likely to be vaccinated for COVID-19. In each ethnoracial group, the majority of respondents were fully vaccinated, but there were some important variations across ethnoracial groups. African American respondents had the lowest percentage of being fully vaccinated (63.01%) while Asians had the highest percentage of being fully vaccinated (85.41%). According to Table 2, the unadjusted model with the ethnoracial measure as the only independent variable shows that in comparison to White respondents, African Americans have a 28.1% reduction in their odds [(0.719−1)*100; p<.01] of reporting being fully vaccinated and "Other" respondents have a 22.9% reduction [(0.771−1)*100; p<.1], while Asians' odds of being fully vaccinated increase by 147.1% [(2.471-1)*100; p<.01]. More males (73.72%) than females (65.76%) reported being fully vaccinated, which parallels the unadjusted model in Table 2 because females' odds of being fully vaccinated are 31.6% lower relative to males [(0.684−1)*100; p<.01]. Lastly, Table 1 illustrates that higher levels of educational attainment were associated with higher likelihood of being fully vaccinated. For example, while 87.99% of respondents with a graduate degree were fully vaccinated, only 56.04% of respondents with a high school diploma reported being fully vaccinated. Similarly, according to Table 2, in comparison to respondents with a high school level of education, all levels above high school have greater odds of being fully vaccinated while the odds for those with less than a high school level of education are reduced almost by half [(0.531−1)*100] (all coefficients are significant at the p<.01-level).

Contrary to Hypothesis 1, older individuals were more likely to be fully vaccinated, which might be a result of the strong public health messages emphasizing the need for these individuals to get vaccinated against COVID-19. The 18–29 age group has the lowest percentage of being fully vaccinated (59.55%) while older age groups, especially the 75+ age group with 85.99%, have notably higher percentages of being fully vaccinated. Shifting over to the unadjusted model in Table 2, with respect to the 30–39 age group, the age groups starting at age 50 or above all have greater odds of being fully vaccinated, while the youngest age group –18–29– have 22.3% odds decrease in being fully vaccinated [(0.777−1)*100; all mentioned coefficients are significant at the p<.01-level].

COVID-19 vaccination status by political parties and religious groups give support to Hypothesis 2: Conservative political background leads to lower COVID-19 vaccinations. Although all groups were more likely to report being fully vaccinated than not, it was to varying degrees. According to the percentage distribution, 86.65% of strong Democrats reported being vaccinated, while 58.54% of strong Republicans reported being vaccinated. For Democrats, stronger party

**Table 2. Odds ratios and exponentials of standard errors from bivariate logistic regression models predicting whether individuals are fully vaccinated for COVID-19, 2023.**

| Independent variables | Odds ratios | Exponential of standard errors | Pseudo $R^2$ |
|---|---|---|---|
| **Sex** | | | 0.006 |
| Male | ref. | | |
| Female | 0.684*** | 0.042 | |
| **Age group** | | | 0.029 |
| 18–29 | 0.777*** | 0.072 | |
| 30–39 | ref. | | |
| 40–49 | 0.929 | 0.089 | |
| 50–64 | 1.442*** | 0.135 | |
| 65–74 | 2.340*** | 0.268 | |
| 75+ | 3.240*** | 0.534 | |
| **Education group** | | | 0.064 |
| Less than high school | 0.531*** | 0.089 | |
| High school | ref. | | |
| Some college | 1.444*** | 0.121 | |
| Associate | 1.695*** | 0.173 | |
| Bachelor | 3.329*** | 0.304 | |
| Graduate | 5.748*** | 0.705 | |
| **Political party** | | | 0.057 |
| Strong Democrat | ref. | | |
| Democrat | 0.489*** | 0.054 | |
| Independent | 0.242*** | 0.025 | |
| Republican | 0.222*** | 0.023 | |
| Strong Republican | 0.218*** | 0.023 | |
| Other party | 0.190*** | 0.052 | |
| **Religious group** | | | 0.027 |
| Evangelical Protestants | ref. | | |
| Other Protestant | 1.457*** | 0.157 | |
| Catholic | 1.959*** | 0.195 | |
| Other | 3.449*** | 0.638 | |
| Agnostic/Atheist | 2.171*** | 0.285 | |
| Nothing in particular | 0.894 | 0.081 | |
| **Race/ethnicity** | | | 0.006 |
| White | ref. | | |
| Hispanic | 0.912 | 0.082 | |
| African American | 0.719*** | 0.067 | |
| Asian | 2.471*** | 0.522 | |
| Other race/ethnicity | 0.771* | 0.114 | |
| **Main source of information about COVID-19 vaccines** | | | 0.060 |
| Centers for Disease Control and Prevention (CDC) | ref. | | |
| Other health sources | 0.426*** | 0.034 | |
| Employer, family, and friends | 0.194*** | 0.022 | |
| News sources and professional organizations | 0.408*** | 0.054 | |
| Other sources | 0.082*** | 0.014 | |

Note: All models have a sample size of 5,240 respondents. Constant terms of each model are not reported.

*(Continued)*

**Table 2.** (Continued)
*Significant at p < 0.1;
**Significant at p < 0.05;
***Significant at p < 0.01.
Source: 2023 Survey about Effective Communication Strategies During the COVID-19 Pandemic.

affiliation is associated with higher rates of vaccination, while for Republicans, stronger party affiliation is minimally associated with lower rates of vaccination. Referring to Table 2's unadjusted model, compared to the reference group –strong Democrat– all other political party categories have lower odds of being fully vaccinated (all coefficients are significant at the p < .01-level). Moreover, for all categories of religious identification, respondents were more likely to report being fully vaccinated than not. However, it is important to note that Evangelical Protestants had one of the smallest percentages to be fully vaccinated against COVID-19 (62.83%) while "Other" has the highest (85.36%). Table 2's unadjusted model adds proof to the possible influence of religious background. An important note is that this variable about religious groups does not measure the strength of religious beliefs or religiosity. All we can assess is whether religious groups are associated with respondents being fully vaccinated for COVID-19. These results indicate that relative to identifying as an Evangelical Protestant, all categories –except for "nothing in particular" – have greater odds of being fully vaccinated (all coefficients are significant at the p < .01-level).

Lastly, individuals who got information on COVID-19 vaccination from the CDC (83.35%) had the highest percentage of all information categories to be fully vaccinated while respondents who reported that "Other sources" such as religious leaders, social media, and union leaders were their main source of information had the lowest percentage of being fully vaccinated (29.22%). Moreover, Table 2's unadjusted model for the main source of information shows that relative to reporting the CDC as their main information source, all other sources are associated with substantially lower odds of a respondent being fully vaccinated, especially those who report "Other sources" with a 91.8% decrease in the odds [(0.082–1)*100]. All coefficients are significant at the p < .01-level. These findings are in line with our Hypothesis 3: Relative to individuals who rely on health experts as their main information source, in particular the CDC, individuals who rely on more informal networks are less likely to be fully vaccinated.

Following the previous bivariate analysis of COVID-19 vaccination by independent variables, we estimated a multivariate logistic regression model to elucidate which factors presented the strongest and statistically significant associations with being fully vaccinated (Table 3). The overall statistical significance of the model is underscored by the Pseudo R-squared of 0.174 and the likelihood ratio chi-squared test of 1,120.84, indicating its robust predictive capacity for vaccination status. Examining the odds ratios for individual predictor variables provides nuanced insights into their effects on vaccination likelihood.

Demographic variables exhibit substantial influence on the chances to be fully vaccinated for COVID-19, as illustrated on Table 3. The patterns in the full model in general follow the broad patterns of Table 2's bivariate analysis. Like in the bivariate analysis, our findings are in line with Hypothesis 1: More vulnerable groups are less likely to be fully vaccinated for COVID-19. However, race/ethnicity groups exhibit interesting patterns related to COVID-19 vaccination that do not entirely fit with this hypothesis. In a multivariate analysis, we would expect that Hispanics have lower levels of vaccination than the White population, because they are a more vulnerable population group. However, our findings indicate that there is no statistically significant difference in the odds of being fully vaccinated among Hispanic compared to White peoples when controlling for all other independent variables. This finding could be related to the Hispanic mortality paradox, i.e., cultural and sociodemographic factors that influence better health behaviors and outcomes among Hispanics, even with their lower socioeconomic status than White people. Asians have a 118.6% increase in their odds of being fully vaccinated for COVID-19 compared to White people, controlled for the other independent variables [(2.186-1)*100; p < .01]. On the other hand, African Americans have roughly a 20% decrease in their odds of being vaccinated relative to White people

**Table 3. Odds ratios and exponentials of standard errors from a multivariate logistic regression model predicting whether individuals are fully vaccinated for COVID-19, 2023.**

| Independent variables | Odds ratios | Exponential of standard errors |
|---|---|---|
| Constant | 4.136*** | 0.707 |
| **Sex** | | |
| Male | ref. | |
| Female | 0.747*** | 0.051 |
| **Age group** | | |
| 18–29 | 0.978 | 0.101 |
| 30–39 | ref. | |
| 40–49 | 1.023 | 0.110 |
| 50–64 | 1.693*** | 0.179 |
| 65–74 | 2.345*** | 0.301 |
| 75+ | 3.245*** | 0.590 |
| **Education group** | | |
| Less than high school | 0.584*** | 0.106 |
| High school | ref. | |
| Some college | 1.222** | 0.112 |
| Associate | 1.470*** | 0.164 |
| Bachelor | 2.551*** | 0.257 |
| Graduate | 3.722*** | 0.501 |
| **Political party** | | |
| Strong Democrat | ref. | |
| Democrat | 0.583*** | 0.069 |
| Independent | 0.365*** | 0.041 |
| Republican | 0.292*** | 0.034 |
| Strong Republican | 0.297*** | 0.035 |
| Other party | 0.305*** | 0.097 |
| **Religious group** | | |
| Evangelical Protestants | ref. | |
| Other Protestant | 1.177 | 0.142 |
| Catholic | 1.567*** | 0.175 |
| Other | 1.785*** | 0.369 |
| Agnostic/Atheist | 1.652*** | 0.250 |
| Nothing in particular | 1.050 | 0.111 |
| **Race/ethnicity** | | |
| White | ref. | |
| Hispanic | 1.163 | 0.120 |
| African American | 0.827* | 0.089 |
| Asian | 2.186*** | 0.497 |
| Other race/ethnicity | 0.886 | 0.149 |
| **Main source of information about COVID-19 vaccines** | | |
| Centers for Disease Control and Prevention (CDC) | ref. | |
| Other health sources | 0.551*** | 0.047 |
| Employer, family, and friends | 0.339*** | 0.042 |
| News sources and professional organizations | 0.477*** | 0.070 |
| Other sources | 0.111*** | 0.021 |

*(Continued)*

**Table 3.** (Continued)

| Independent variables | Odds ratios | Exponential of standard errors |
|---|---|---|
| Likelihood Ratio Chi-squared Test (df = 29) | 1,120.84*** | |
| Pseudo R-squared | 0.174 | |
| Observations | 5,240 | |

Note:

*Significant at p < 0.1;

**Significant at p < 0.05;

***Significant at p < 0.01.

Source: 2023 Survey about Effective Communication Strategies During the COVID-19 Pandemic.

[(.827-1)*100; p < .1)], which emphasizes another vulnerable situation this group experiences in the population. Unlike in the bivariate analysis, the results for the "Other" ethnoracial category are no longer even marginally significant.

There is additional support for Hypothesis 1 when focusing on the variables "sex" and "education group". According to the adjusted model, females' odds of being vaccinated are approximately 25% lower [(.747-1)*100; p < .01], compared to males, suggesting a gender-based discrepancy in vaccination access and decisions. Our findings support existing research on COVID vaccines that has pointed to the gender discrepancy in vaccination rates [58,59]. These studies indicate that, in contrast to men who are more likely to minimize health risk of COVID 19 and believe in conspiratorial theories, women's vaccine hesitancy is a result of perceived fear of side effects of the vaccine [58]. Moreover, educational attainment manifests as a strong predictor in the adjusted model, with a discernible trend of higher odds of COVID-19 vaccination associated with increased educational levels. For example, compared to people with a high school degree, those with a bachelor's degree have an 155.1% increase [(2.551-1)*100] in their odds of being fully vaccinated for COVID-19 while those with a graduate degree have a 272.2% increase [(3.722-1)*100]. Both coefficients are significant at the p < .01-level.

Not in line with Hypothesis 1, individuals in the 50–64, 65–74, and 75 + age groups display increased odds of vaccination, all statistically significant at the p < .01-level. For instance, those age 65–74 have a 134.5% increase [(2.345-1)*100] in their odds of being fully vaccinated for COVID-19, compared to those ages 30–39, controlling for the other independent variables. As noted before, the more dangerous nature of the COVID-19 virus on older individuals might have influenced their vaccination status. Lastly, although the direction of the coefficients are comparable to the bivariate analysis, there is no statistically significant difference between ages 18–29 and 30–39 according to the multivariate analysis.

Notably, as expected by Hypothesis 2, political leanings and religious beliefs demonstrate significant associations with vaccination likelihood in Table 3. Our findings underscore the intricate interplay of cultural, political, and social factors in shaping vaccination decisions. As seen in the bivariate analysis, estimates for political parties according to the multivariate analysis indicate that individuals who self-classify as strong Democrats are more likely to be fully vaccinated for COVID-19, significant at the p < .01-level and controlling for the other independent variables. For instance, compared to strong Democrats, the odds for Democrats decrease by 41.7% [(.583-1)*100] to be vaccinated. This likelihood reaches the lowest values for Republicans [a reduction by 70.8%; (.292-1)*100] and strong Republicans [a reduction by 70.3%; (.297-1)*100]. These findings emphasize that political leanings have strong associations with how individuals faced the pandemic, which includes getting vaccinated for COVID-19.

In terms of religious groups, Evangelical Protestants have the lowest chances of being fully vaccinated for COVID-19 among all religious groups. For instance, Catholics are 56.7% more likely [(1.567–1)*100] to be fully vaccinated than Evangelical Protestants, controlling for the other independent variables. The increase in the likelihood reaches 78.5% [(1.785–1)*100] among individuals who fall in the "Other" category and a 65.2% increase [(1.652–1)*100] among

**Table 4. Percentage distribution of respondents by source of information about COVID-19 vaccines and race/ethnicity, as well as percentage of those fully vaccinated in each sub-group, 2023.**

| Main source of information about COVID-19 vaccines | Race/ethnicity | | | | | |
|---|---|---|---|---|---|---|
| | White | Hispanic | African American | Asian | Other | Total |
| **Centers for Disease Control and Prevention (CDC)** | | | | | | |
| Sample size | 1,112 | 196 | 172 | 73 | 63 | 1,616 |
| Column percentage | 31.17 | 28.32 | 29.45 | 39.46 | 29.72 | 30.84 |
| Percentage fully vaccinated | 84.17 | 82.14 | 76.74 | 95.89 | 76.19 | 83.35 |
| **Other health sources** | | | | | | |
| Sample size | 1,790 | 360 | 274 | 83 | 98 | 2,605 |
| Column percentage | 50.18 | 52.02 | 46.92 | 44.86 | 46.23 | 49.71 |
| Percentage fully vaccinated | 69.11 | 64.17 | 61.31 | 85.54 | 67.35 | 68.06 |
| **Employer, family, and friends** | | | | | | |
| Sample size | 320 | 76 | 63 | 5 | 21 | 485 |
| Column percentage | 8.97 | 10.98 | 10.79 | 2.70 | 9.91 | 9.26 |
| Percentage fully vaccinated | 49.38 | 59.21 | 41.27 | 20.00 | 42.86 | 49.28 |
| **News sources and professional organizations** | | | | | | |
| Sample size | 219 | 39 | 50 | 19 | 14 | 341 |
| Column percentage | 6.14 | 5.64 | 8.56 | 10.27 | 6.60 | 6.51 |
| Percentage fully vaccinated | 64.84 | 84.62 | 68.00 | 68.42 | 50.00 | 67.16 |
| **Other sources** | | | | | | |
| Sample size | 126 | 21 | 25 | 5 | 16 | 193 |
| Column percentage | 3.53 | 3.03 | 4.28 | 2.70 | 7.55 | 3.68 |
| Percentage fully vaccinated | 27.78 | 14.29 | 32.00 | 60.00 | 43.75 | 29.02 |
| **Total** | | | | | | |
| Sample size | 3,567 | 692 | 584 | 185 | 212 | 5,240 |
| Column percentage | 100.00 | 100.00 | 100.00 | 100.00 | 100.00 | 100.00 |
| Percentage fully vaccinated | 70.31 | 68.35 | 63.01 | 85.41 | 64.62 | 69.54 |

Note: Percentage of respondents fully vaccinated for COVID-19 are reported between parentheses.

Source: 2023 Survey about Effective Communication Strategies During the COVID-19 Pandemic.

Agnostic/Atheist individuals. All mentioned coefficients are significant at the $p < .01$-level. However, some religious affiliations ("Other Protestants" and nothing in particular) have higher odds of being fully vaccinated than Evangelical Protestants, but their results are not statistically significant. Besides the change in statistical significance for the "Other Protestant" coefficient, the major patterns of religious affiliation in the multivariate analysis are comparable to the results of the bivariate analysis, which suggests that religious affiliation –like political affiliation– each capture distinct phenomena that are strongly correlated with being fully vaccinated.

Findings also indicate strong and statistically significant associations between the main source of information about COVID-19 vaccines and being fully vaccinated. Individuals whose main source of information is the CDC have the highest chances to be fully vaccinated, controlling for the other independent variables (Table 3). Relative to reporting the CDC as the main source of information, the lowest odds ratios of being fully vaccinated are for individuals whose main sources of vaccine information comes from "Other sources" such as religious leaders, social media, and union leaders [a decrease by 88.9%; (.111-1)*100] and it is statistically significant at the $p < .01$-level. Other interesting findings indicate that the odds for being fully vaccinated among individuals who mainly receive information from their employer or family and friends is reduced by 66.1% [(.339-1)*100; $p < .01$] compared to the CDC category, while the category with the second-highest odds

**Table 5. Odds ratios and exponentials of standard errors from a multivariate logistic regression model predicting whether individuals are fully vaccinated for COVID-19, 2023.**

| Independent variables | Odds ratios | Exponential of standard errors |
|---|---|---|
| **Race/ethnicity interacted with main source of information about COVID-19 vaccines** | | |
| **White** | | |
| Centers for Disease Control and Prevention (CDC) | ref. | |
| Other health sources | 0.545*** | 0.057 |
| Employer, family, and friends | 0.313*** | 0.048 |
| News sources and professional organizations | 0.367*** | 0.067 |
| Other sources | 0.101*** | 0.024 |
| **Hispanic** | | |
| Centers for Disease Control and Prevention (CDC) | 1.054 | 0.228 |
| Other health sources | 0.550*** | 0.083 |
| Employer, family, and friends | 0.562** | 0.150 |
| News sources and professional organizations | 1.431 | 0.677 |
| Other sources | 0.057*** | 0.036 |
| **African American** | | |
| Centers for Disease Control and Prevention (CDC) | 0.720 | 0.153 |
| Other health sources | 0.423*** | 0.069 |
| Employer, family, and friends | 0.253*** | 0.073 |
| News sources and professional organizations | 0.609 | 0.204 |
| Other sources | 0.113*** | 0.051 |
| **Asian** | | |
| Centers for Disease Control and Prevention (CDC) | 4.137** | 2.507 |
| Other health sources | 1.357 | 0.459 |
| Employer, family, and friends | 0.072** | 0.081 |
| News sources and professional organizations | 0.616 | 0.325 |
| Other sources | 0.254 | 0.252 |
| **Other race/ethnicity** | | |
| Centers for Disease Control and Prevention (CDC) | 0.634 | 0.209 |
| Other health sources | 0.488*** | 0.122 |
| Employer, family, and friends | 0.272*** | 0.131 |
| News sources and professional organizations | 0.379* | 0.216 |
| Other sources | 0.221*** | 0.121 |
| Likelihood Ratio Chi-squared Test (df=45) | 1,148.55*** | |
| Pseudo R-squared | 0.178 | |
| Observations | 5,240 | |

Note: This model is controlled for sex, age group, education group, political party, and religious group.

*Significant at $p < 0.1$;

**Significant at $p < 0.05$;

***Significant at $p < 0.01$.

Source: 2023 Survey about Effective Communication Strategies During the COVID-19 Pandemic.

**Table 6. Odds ratios and exponentials of standard errors from a multivariate logistic regression model for each race/ethnicity category predicting whether individuals are fully vaccinated for COVID-19, 2023.**

| Independent Variables | Race/ethnicity | | | | |
|---|---|---|---|---|---|
| | White | Hispanic | African American | Asian | Other |
| Constant | 3.962*** | 17.372*** | 2.744*** | 5.928 | 1.742 |
| | (0.847) | (9.615) | (1.048) | (10.469) | (1.497) |
| **Sex** | | | | | |
| Male | ref. | ref. | ref. | ref. | ref. |
| Female | 0.708*** | 0.650** | 0.843 | 1.284 | 0.922 |
| | (0.061) | (0.125) | (0.163) | (0.716) | (0.327) |
| **Age group** | | | | | |
| 18–29 | 0.956 | 1.097 | 0.853 | 2.767 | 0.929 |
| | (0.128) | (0.283) | (0.236) | (2.179) | (0.516) |
| 30–39 | ref. | ref. | ref. | ref. | ref. |
| 40–49 | 0.986 | 0.863 | 1.214 | 1.922 | 2.461 |
| | (0.134) | (0.252) | (0.344) | (1.895) | (1.530) |
| 50–64 | 1.732*** | 1.214 | 1.839** | 3.669 | 1.974 |
| | (0.229) | (0.352) | (0.540) | (3.291) | (1.072) |
| 65–74 | 2.672*** | 0.790 | 2.690*** | 5.944 | 1.444 |
| | (0.422) | (0.340) | (1.022) | (6.443) | (0.801) |
| 75+ | 3.512*** | 1.802 | 4.579* | 0.996 | 2.522 |
| | (0.734) | (1.458) | (3.629) | (1.350) | (2.010) |
| **Education group** | | | | | |
| Less than high school | 0.589** | 0.541 | 0.597 | dropped[1] | 0.152 |
| | (0.145) | (0.235) | (0.251) | | (0.179) |
| High school | ref. | ref. | ref. | ref. | ref. |
| Some college | 1.178 | 1.439 | 1.121 | 3.052 | 1.617 |
| | (0.136) | (0.354) | (0.287) | (2.544) | (0.745) |
| Associate | 1.493*** | 1.915** | 1.213 | 6.650* | 0.918 |
| | (0.210) | (0.583) | (0.359) | (7.327) | (0.512) |
| Bachelor | 2.508*** | 3.396*** | 2.027** | 8.726** | 5.724*** |
| | (0.305) | (1.071) | (0.590) | (7.850) | (3.563) |
| Graduate | 4.133*** | 3.554*** | 4.339*** | 5.343* | 2.498 |
| | (0.680) | (1.467) | (2.277) | (5.260) | (1.528) |
| **Political party** | | | | | |
| Strong Democrat | ref. | ref. | ref. | ref. | ref. |
| Democrat | 0.631*** | 0.261*** | 0.615* | 3.356 | 0.776 |
| | (0.098) | (0.097) | (0.160) | (3.714) | (0.465) |
| Independent | 0.360*** | 0.171*** | 0.498** | 2.525 | 0.362* |
| | (0.053) | (0.061) | (0.138) | (2.325) | (0.207) |
| Republican | 0.296*** | 0.119*** | 0.398*** | 1.936 | 0.355* |
| | (0.044) | (0.047) | (0.127) | (1.796) | (0.204) |
| Strong Republican | 0.274*** | 0.249*** | 0.659 | 0.743 | 0.246** |
| | (0.040) | (0.109) | (0.213) | (0.851) | (0.152) |
| Other party | 0.267*** | 0.368 | 0.074** | dropped[1] | 0.177* |
| | (0.116) | (0.275) | (0.091) | | (0.174) |

*(Continued)*

**Table 6.** (Continued)

| Independent Variables | Race/ethnicity | | | | |
|---|---|---|---|---|---|
| | White | Hispanic | African American | Asian | Other |
| **Religious group** | | | | | |
| Evangelical Protestants | ref. | ref. | ref. | ref. | ref. |
| Other Protestant | 1.407** | 0.327*** | 1.027 | dropped[1] | 1.010 |
| | (0.206) | (0.135) | (0.316) | | (0.608) |
| Catholic | 1.766*** | 1.046 | 1.189 | 0.123 | 1.667 |
| | (0.240) | (0.349) | (0.422) | (0.177) | (1.070) |
| Other | 2.041*** | 1.086 | 0.951 | 0.402 | 3.075 |
| | (0.514) | (0.835) | (0.561) | (0.636) | (3.239) |
| Agnostic/Atheist | 1.934*** | 0.740 | 1.206 | 0.184 | 2.179 |
| | (0.349) | (0.345) | (0.710) | (0.278) | (1.709) |
| Nothing in particular | 1.105 | 0.814 | 0.966 | 0.129 | 1.190 |
| | (0.143) | (0.286) | (0.254) | (0.179) | (0.676) |
| **Main source of information about COVID-19 vaccines** | | | | | |
| Centers for Disease Control and Prevention (CDC) | ref. | ref. | ref. | ref. | ref. |
| Other health sources | 0.554*** | 0.424*** | 0.590** | 0.259* | 0.990 |
| | (0.059) | (0.104) | (0.140) | (0.192) | (0.439) |
| Employer, family, and friends | 0.327*** | 0.437** | 0.373*** | dropped[1] | 0.480 |
| | (0.050) | (0.146) | (0.128) | | (0.304) |
| News sources and professional organizations | 0.368*** | 0.995 | 0.828 | 0.080** | 0.626 |
| | (0.068) | (0.518) | (0.315) | (0.079) | (0.461) |
| Other sources | 0.103*** | 0.046*** | 0.156*** | 0.028** | 0.438 |
| | (0.025) | (0.032) | (0.077) | (0.040) | (0.291) |
| Likelihood Ratio Chi-squared Test | 847.95*** (df = 25) | 155.08*** (df = 25) | 102.20*** (df = 25) | 32.73** (df = 21) | 47.76*** (df = 25) |
| Pseudo R-squared | 0.195 | 0.180 | 0.133 | 0.247 | 0.173 |
| Observations | 3,567 | 692 | 584 | 162 | 212 |

[1]Among Asians, the statistical software automatically dropped some individuals from the analysis, because in specific categories all individuals were fully vaccinated (predicting success perfectly). A total of 23 respondents were omitted: three respondents with less than high school; two respondents with other party affiliation; 14 respondents in other Protestant religious group; and four respondents whose main source of information about COVID-19 vaccines was from employer, family, and friends. As a result, the sample size of Asians dropped from 185 to 162 individuals.

Note: Exponential of standard errors are reported between parentheses.

*Significant at p < 0.1;

**Significant at p < 0.05;

***Significant at p < 0.01.

Source: 2023 Survey about Effective Communication Strategies During the COVID-19 Pandemic.

ratios of being fully vaccinated are those who rely on "Other health sources" as their main source of information. However, the difference between this category and the reference category is statistically significant at the p < .01-level Overall, these findings –which parallel the bivariate results– highlight that reliance on informal networks such as religious leaders, social media, and personal contacts to receive information about vaccines have the lowest effects on individuals being fully vaccinated for COVID-19, as expected by Hypothesis 3.

Table 4 further emphasizes the combined impact of ethnoracial background and main source of information about COVID-19 vaccines by showing the percentage distribution of main sources of information among White, Hispanic, African

American, Asian, and "Other" respondents as well as the percentage of those being fully vaccinated by ethnoracial background and main source combination. Across all ethnoracial categories, the most common primary sources of information about vaccines are "Other health sources" (44.86% among Asians to 52.02% among Hispanics) and the CDC (28.83% among Hispanics to 39.42% among Asians). In turn, more than half of respondents who selected one of these sources as their main source of information, across all ethnoracial categories, report being fully vaccinated. However, there are notable discrepancies worth discussing. For example, just as Asians seem to be at a statistically significant advantage in comparison to White respondents in the adjusted model, this advantage is evident in Table 4's descriptive statistics. 95.89% and 85.54% of Asian respondents who report the CDC or "Other health sources" as their main source, respectively, were fully vaccinated. Alternatively, among respondents who selected the CDC as their main source, African Americans and respondents belonging to other ethnoracial groups had the lowest rates of being fully vaccinated (76.74% and 76.19%, respectively). Similarly, among respondents whose response falls within the category of "Other health sources", besides Asians, all other groups report full vaccination rates below 70% and African Americans had the lowest percentage (61.31%).

Less common main sources of information for COVID-19 vaccination further highlight differences across ethnoracial groups. For example, employer or family and friends was the third most common category among all groups minus Asians. A considerable percentage of people indicate that their main source of information about COVID-19 vaccines is employer or family and friends: 8.97% among White people and 10.98% among Hispanics and African Americans. For Asians (n = 5; 2.70%) and "Other" race/ethnicity (n = 21; 9.91%) the sample sizes are small. Due to the smaller sample sizes among the latter groups, vaccination rates should be interpreted with caution. As the adjusted model in Table 3 suggests, vaccination rates for this source category are consistently lower across all ethnoracial groups, especially for Asians (20.00%) and African Americans (41.27%). The only ethnoracial group to have a vaccination rate over 50% in this source category is Hispanics (59.21%). Lastly, for White, Hispanics, and African American respondents, "Other sources" (non-health related) are the least commonly reported main source, and the percentage of those being fully vaccinated are especially low among these mentioned ethnoracial groups (14.29% to 27.78%).

Table 5 shows the results of one multivariate logistic regression model that has interaction effects between race/ethnicity and information source for COVID-19 vaccination as a set of dummies. This model also controls for sex, age group, education group, political party, and religious group. The interaction effects in an adjusted model enhance our understanding of certain patterns evident in Table 4. For example, Asian respondents who select the CDC as their main source of information have over a 300% increase [(4.137-1)*100; p < .05] in the odds of being fully vaccinated relative to White people who select the CDC as their main source of information. Moreover, no other ethnoracial group who report the CDC as their main information source has a statistically significant difference relative to White people who selected the CDC. Second, Asians who report some form of "Other health sources" as their main information source are the only ethnoracial group to not have a statistically significant difference in comparison to White respondents who report the CDC as their main source while other groups such as Hispanics and African American who select "Other health sources" as their main information source are at a significant disadvantage. The information category "Other sources" has a similar pattern. Third, for those with employer or family and friends as their main information source, White people's odds of being fully vaccinated decrease by 68.7% [(0.313–1)*100; p < .01], Hispanics decrease by 43.8% [(0.562–1)*100; p < .05], and African Americans decrease by 74.7% [(0.253–1)*100; p < .01], compared to White respondents who select the CDC as their main information source. Lastly, for the response category "news sources and professional organizations", there is a range in terms of whether certain ethnoracial groups have significantly lower odds of being fully vaccinated in comparison to White respondents who report the CDC as their main source of information.

Table 6 shows the results for separate multivariate logistic regression models by ethnoracial background. These models show an even more nuanced analysis of each group since running a model by ethnoracial background is equivalent to interacting race/ethnicity with all independent variables. Some categories for Asians were automatically dropped by the

statistical software due to small sample size. In this case, the dropped category is added to the reference category. For instance, since the category of Asians with less than high school was dropped, the remaining odds ratios (some college, associate degree, bachelor, and graduate) have to be interpreted as a comparison to less than high school and high school combined.

However, we focus the analysis on findings that were statistically significant for the reported main source of information about COVID-19 vaccination (reference category: CDC). More specifically, given the lack of statistically significant differences in the odds ratios for main source of information for Asian and other race/ethnicity, the written analysis focuses on White, African American, and Hispanic respondents with particular attention to associations between the main explanatory variable and likelihood of being fully vaccinated.

This last set of results highlight multiple similarities among White, Hispanic, and African American respondents. For example, those who report that their employer or family and friends are their main information source have considerably lower chances to be fully vaccinated, compared to those with CDC as their main information source in each race/ethnicity category: a decrease by 67.3% [(0.327–1)*100; $p < .01$] among White people, 56.3% [(0.437–1)*100; $p < .05$] among Hispanics, and 62.7% [(0.373–1)*100; $p < .01$] among African Americans. The response categories "Other health sources" and "Other sources" have similar patterns. For "Other sources", the evidence is abundantly clear that compared to those with CDC as main information source in each race/ethnicity category, this category is associated with a dramatic decrease in the odds of being fully vaccinated ($p < .01$ for all three coefficients across the models). The odds decrease by 89.7% [(0.103–1)*100] for White people, 95.4% [(0.046–1)*100] for Hispanics, and 84.4% [(0.156–1)*100] for African Americans. The most notable difference is that among respondents who select news sources and professional organizations as their main source of information, only White respondents showed a statistically significant difference to respondents of the same ethnoracial group who selected the CDC. More specifically, there is a 63.2% decrease in the odds of being fully vaccinated [(0.368–1)*100; $p < .01$].

## Final considerations and future directions

In developing, introducing, and distributing a new vaccine, experts often work against the pressures exerted by costs and time. Once an effective and accessible vaccine is in place, the major obstacle in its distribution remains the vaccine's acceptability. Vaccine acceptability has therefore been the leading indicator for the overall success of vaccination programs [60]. In an age where anti-vaccination attitudes and behavior along with misconceptions about vaccination are rampant [13], increasing vaccine uptake becomes an even more challenging task for policy makers. The World Health Organization therefore cites vaccine refusal and hesitancy in its list of top threats to global public health [50]. Providing insights on the factors influencing the behavior towards COVID-19 vaccines, this study proposes that policies geared toward increasing the acceptance rate of vaccines should be tailored to community characteristics, in particular to political and religious beliefs, and race/ethnicity-related characteristics of a community. It further suggests that information sources play a particularly crucial role in tailoring vaccine campaigns to specific communities.

Our results reveal that vulnerable populations of women, African Americans and those who did not have a college degree were less likely to be vaccinated. Moreover, those younger than 50 years of age exhibited lower levels of vaccination compared to those who are older than 50. These findings confirm the Health Belief Model (HBM) that suggests that people are more likely to engage in health-promoting behavior if they believe they are at a greater health risk and that taking appropriate action will reduce that risk [61,62]. It is well documented that older adults are more susceptible to COVID -19 and are also more likely to experience the pandemic severely. Data indicates that 3 out of 4 deaths caused by COVID-19 are deaths of older adults [63]. Thus, the benefits of vaccination outweigh the barriers to action for older adults. By presenting that old age drives vaccine uptake, our findings indicate that HBM is a useful model for understanding why older adults are more likely to get vaccinated. The study thus reaffirms existing proposals [61] that suggest that public

health messaging should highlight the risks that arise from the pandemic and the benefits of vaccination in preventing these risks.

At the same time our findings challenge those who made claims against significant racial or ethnic differences in attitudes toward COVID-19 [64]. Specifically, while for all ethnoracial groups, most of the respondents were fully vaccinated, according to the adjusted model, Asians had a statistically significant increase in their odds of being fully vaccinated compared to Whites, and African Americans had a substantial decrease in their odds of being fully vaccinated in relation to Whites. The study thus identified women, African Americans, those who have less access to education, and those who are younger than 50 as communities that are in need of community tailored vaccine programs that speak to each community's unique conditions and vulnerabilities. It suggests that public health messages that promote vaccination should be tailored to these groups' specific beliefs regarding risks.

In line with much of existing research [39,43] we found that Democrats, especially strong Democrats, were more likely to be vaccinated compared to Republicans. These results emphasized the strong association between political identity and vaccine behavior during the COVID-19 pandemic. They further pointed at the importance of recognizing political polarization as an impediment to the success of vaccination campaigns. Sociologists have previously shown that in politically polarized environments an important strategy in mobilizing people behind a campaign is depoliticizing the debate through using scientific frames [65]. When there is political polarization over scientific conclusions such as the need to get the COVID-19 vaccine that strategy however becomes harder to implement. One way to deal with this barrier would be, when shaping campaigns, health authorities and policy makers should recognize and utilize community specific values and norms.

During the COVID-19 pandemic we have seen that some highly religious groups ascribe spiritual forces, both demonic and divine, to the pandemic and the development of vaccines [66]. While those that had demonic vaccine attributions had lower vaccinations rates and greater vaccine hesitancy, those that attributed the development of a vaccine to divine forces had higher vaccine rates for themselves and their children, more positive attitudes towards vaccines, and were more likely to see COVID-19 as a threat. They tended to view vaccines as a way to love their neighbors, and thus live out their religious mandates. While this is only one example, it does show the importance of recognizing community-specific systems of values and how governments can leverage prevailing norms of behavior to promote vaccination.

Our own findings on information sources suggest that another major way of overcoming the rampant political polarization in American society regarding science and vaccine uptakes would be the establishment of alliances with primary information sources in targeted communities. Our findings indicate strong and statistically significant associations between the main source of information about COVID-19 vaccines and being fully vaccinated. Specifically, those individuals whose main sources of vaccine information were informal sources such as employers, family and friends, religious leaders, social media, and union leaders were less likely to report being fully vaccinated. In contrast, those who followed health experts such as the CDC, FDA, hospital system websites, and primary care providers for information on COVID-19 were more likely to get vaccinated. These findings assert that the mediator of vaccination campaigns is at least as important as the message of the vaccination campaign. To diminish misperceptions and generate trust to vaccination programs, we need greater collaboration between the CDC and community specific sources of information. These efforts may involve partnering with individuals that those in more niche, potentially anti-vaccine populations trust [52]. These could include faith leaders, but also doctors and trusted health professionals.

Overall, identifying primary sources of information in given communities, increasing the partnerships and frequency of communication between the CDC and community specific information sources, and tailoring vaccination campaigns to the needs and perceptions of vulnerable communities emerge from our findings as three major strategies in motivating vaccine-hesitant individuals towards vaccine acceptance. Acknowledging and addressing these factors is imperative for the development of targeted public health interventions aimed at promoting widespread COVID-19 vaccination access and acceptance.

Communication strategies that target behavioral change are at the center of contemporary community-based public health interventions [67]. One study [68] specifically emphasizes evidence-based communication strategies, noting that they are essential when

dealing with community members in order to control COVID-19 vaccine-related misinformation and to ensure large public benefits. Building on this insight, our study offers a foundation for tailoring vaccine campaigns to communities. Based on our findings, we propose that information sources constitute a primary means of tailored vaccine campaigns. While communication of content is important for vaccine acceptance–and not just through short-term campaigns but also through long-term communicative approaches [69], our results indicate that vaccine campaigns reach relevant audiences only when they are distributed by trusted sources. By highlighting possible mechanisms through which public health messages reach the target populations, we offer a framework that can help tailor communication messages to communities and make them more efficient. These strategies can then increase vaccination uptake, saving lives and valuable resources during the current and future pandemics led by emerging infectious diseases.

A key limitation of our correlational study is that we cannot make any causal inferences as to which factors are driving vaccination behavior. Another limitation is that the data are collected retrospectively and thus rely on respondents' ability to recall accurately 1) if they have received a COVID-19 vaccination, and if so, 2) did they complete the COVID-19 vaccination schedule. While we do make assumptions about respondents' abilities to correctly remember their vaccination status, collecting data on vaccination behavior after the peak of the pandemic grants us a fuller picture of people who vaccinated earlier on as vaccines were first released for public use as well later after the initial wave of vaccinations.

Future research should analyze predictors of vaccination by accessing data that may capture vaccination behavior more accurately, such as respondents' health records. Although this study features a large sample size that speaks to the broad U.S. population as well as a thorough, detailed analysis of several factors that can impact vaccination behavior, an insightful avenue of future research is to reconduct this analysis using another form of vaccination data.

Further, acceptance of a new vaccine is often addressed as a matter of psychological behavior where people feeling fearful, or distrustful towards the vaccine show unwillingness toward taking the vaccine [70]. Indeed, during the COVID-19 pandemic, scholars have shown that public trust in the vaccine to be safe and effective after administration was the strongest forecaster of COVID-19 vaccine uptake intention. While emotions certainly play a key role in motivating one's actions towards a new vaccine, social sciences remind us that emotions are socially produced [71–73]. The framing of information and how we receive the information play key roles in giving direction to our emotions vis-a-vis vaccines. Our findings also call for further scholarship that understands emotions toward vaccines in the context of individuals' community memberships. Afterall, in tailoring vaccination campaigns to specific vulnerabilities and needs of communities the task ahead will be enhancing community members' trust to the vaccine, its developers and distributors.

## Supporting information

**S1 Database. Database in excel.**
(XLSX)

**S2 Database. Database in DTA (Stata).**
(DTA)

**S3 Codes. Codes in TXT.**
(TXT)

**S4 Codes. Codes in DO (Stata).**
(DO)

## Author contributions

**Conceptualization:** Defne Över, Emilce Santana, Ernesto F. L. Amaral, Chaitanya Lakkimsetti.

**Data curation:** Defne Över, Emilce Santana, Ernesto F. L. Amaral, Chaitanya Lakkimsetti.

**Formal analysis:** Ernesto F. L. Amaral.

**Funding acquisition:** Defne Över, Emilce Santana, Ernesto F. L. Amaral, Chaitanya Lakkimsetti.

**Investigation:** Defne Över, Emilce Santana, Ernesto F. L. Amaral, Chaitanya Lakkimsetti.

**Methodology:** Defne Över, Emilce Santana, Ernesto F. L. Amaral, Chaitanya Lakkimsetti, Anna Estelle Kelley, Dulce Angelica Espinoza.

**Project administration:** Defne Över, Emilce Santana, Ernesto F. L. Amaral, Chaitanya Lakkimsetti.

**Supervision:** Defne Över, Emilce Santana, Ernesto F. L. Amaral, Chaitanya Lakkimsetti.

**Writing – original draft:** Defne Över, Emilce Santana, Ernesto F. L. Amaral, Chaitanya Lakkimsetti, Anna Estelle Kelley, Dulce Angelica Espinoza.

**Writing – review & editing:** Defne Över, Emilce Santana, Ernesto F. L. Amaral, Chaitanya Lakkimsetti, Anna Estelle Kelley, Dulce Angelica Espinoza.

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
