## [Decision Letter · Decision Letter 0]

22 Oct 2024

PONE-D-24-37455A Comprehensive Analysis of COVID-19 Vaccination Behavior: The Influence of Religion, Information Sources, Political Leanings, and Demographic FactorsPLOS ONE

Dear Dr. Over,

Thank you for submitting your manuscript to PLOS ONE. After careful consideration, we feel that it has merit but does not fully meet PLOS ONE’s publication criteria as it currently stands. Therefore, we invite you to submit a revised version of the manuscript that addresses the points raised during the review process. Specifically, the following three publication criteria have not been fully met and need to be addressed in your revised manuscript:  "Experiments, statistics, and other analyses are performed to a high technical standard and are described in sufficient detail." "Conclusions are presented in an appropriate fashion and are supported by the data." "The article is presented in an intelligible fashion and is written in standard English." This study has the potential to contribute to the broad audience of our journal. However, several significant issues within the manuscript must be addressed before it can be considered for publication. The peer reviewers have provided detailed and constructive feedback, which I believe will greatly enhance the quality and clarity of your work.

We look forward to receiving your revised manuscript.

Kind regards,

Weijun Yu, Ph.D., M.D., M.S.

Academic Editor

PLOS ONE

2.  Please amend your list of authors on the manuscript to ensure that each author is linked to an affiliation. Authors’ affiliations should reflect the institution where the work was done (if authors moved subsequently, you can also list the new affiliation stating “current affiliation:….” as necessary).

3.  Please include a caption for table 1 and 2.

6.  We are unable to open your Supporting Information file [Codes_in_Stata.do and Database_in_Stata.dta]. Please kindly revise as necessary and re-upload.

Reviewers' comments:

Reviewer's Responses to Questions

**Comments to the Author**

1. Is the manuscript technically sound, and do the data support the conclusions?

Reviewer #1: No

Reviewer #2: No

2. Has the statistical analysis been performed appropriately and rigorously? 

Reviewer #1: No

Reviewer #2: N/A

3. Have the authors made all data underlying the findings in their manuscript fully available?

Reviewer #1: Yes

Reviewer #2: Yes

4. Is the manuscript presented in an intelligible fashion and written in standard English?

Reviewer #1: Yes

Reviewer #2: No

5. Review Comments to the Author

Reviewer #1: The description of the methodology for conducting the survey is clearly inadequate.

The organization that conducted the survey, Cloud Research, is named but no information is given about the organization or references to other research conducted by/with the organization is given.

That written informed consent was obtained suggests that the survey was done through face to face interviewing, but this is not clear. Mail interviewing might also have been used.

How subjects were selected and contacted is not specified. Was this a household survey?

The response rate and the methods for determining the response rate are not presented.

Table 2 presents bivariate analyses of the independent variables with the vaccination outcome, but the text describes these relationships as “controlling for other variables. Multivariable logistic results are not included in Table 2.

The data analysis could have included some form of data reduction, such as factor analysis for the many independent variables.

While theory is not required for PLoS One papers, the Health Belief Model would seem to fit these data quite well. It would explain why older subjects were more likely to get vaccinated, which is not explained in the current Discussion.

Reviewer #2: Thank you for the opportunity to review this article. While this topic of research is extremely valuable in the development of tailored future vaccination campaigns, especially considering the diverse demographic composition of the United States, the current paper has several limitations and theoretical shortcomings as discussed below.

First, the paper in its current state has several grammatical errors and needs to be heavily edited to ensure succinct sentences for scientific clarity. For ex, Page 1 - paragraph 1, there are several punctuations missing (i.e., missing comma in "During the COVID-19 pandemic vaccines emerged as a...")

Second, the paper has several instances of poor sentence construction -

1. "In the course of the pandemic, reports have shown that the vaccines also prevented disease spread suggesting that vaccinated individuals would not transmit the disease." This can be changed to "Medical reports published during the pandemic established the role of vaccines in limiting the spread of the virus").

2. Page 2, "Recognizing that

factors influencing vaccine behavior vary depending on time, place, and vaccines [9, 10, 11], researchers trace the variation in vaccine acceptance to historical, socio-cultural, or political experiences of communities with vaccine administrators, to individual and group level perceptions of the vaccine, to the characteristics of the vaccine and vaccination process." Or is a misplaced word in this sentence and overall this is a difficult sentence to read and clearly understand.

3. Page 3, paragraph 2's last sentence refers to something called a "common's decision", are the authors referring to individual vaccine decisions being impacted by social norms/a community's opinion? If so, this is not clearly communicated.

4. Page 3, last paragraph mentions community characteristics but do not specify what these characteristics are.

5. Page 5, paragraph 2's last sentence reads as, "... women have been getting more vaccines than men", can be re-written to "...women having higher vaccination rates than men."

6. Page 6, paragraph 2's first sentence is very long and can be broken up into 2 separate sentences to make it more readable.

Third, citations have not been consistently included -

1. Page 1 - paragraph 1, "At the same time, with intention to get vaccinated is often associated with higher levels of education...").

2. Page 6, last paragraph's first sentence is missing a citation.

Fourth, there are several theoretical and logical gaps in this paper -

1. In its current state, this paper does not establish a clear rationale for how this study adds value to existing research on the topic. It appears that there already exists a lot of data examining COVID-19 vaccination behaviors among various demographic groups in the US as discussed by the authors. How then, is the current study a value addition on the topic?

2. The presentation of a summary of the findings on page 4 seems misplaced since a strong argument for the current study has not yet been presented. One of the findings included on page 4 mentions that "religious beliefs" among Evangelical Protestants are linked to lower vaccination rates. Based on the data collected, as mentioned by the authors, there was no measure of the strength of religious beliefs collected, rather data was collected about what religion one identified with or followed. Therefore, this conclusion is not representative of the data and the findings.

3. Page 6, paragraph 1 presents stats from other studies inconsistently. It would be better to include these statistics consistently for differences between genders, older vs. younger adults, education and income, rather than just reporting them for urban vs. rural areas.

4. It would serve well to end the section of the theoretical background by (a) making a case for the current study, (b) including the motivation behind studying this data, (c) whether it adds any unique perspectives to the topic in addition to the research that already exists; (d) if this is a replication of relationships already studied?; (e) how it would aid in the development of vaccination campaigns, etc.

5. The presentation of hypothesis 1 on page 8 suggests a mediation relationship (i.e., being a member of vulnerable populations leading to the experience of greater obstacles which impacts their likelihood to be vaccinated). However, a mediation relationship was not tested. It would be better to indicate these relationships only when discussing the theoretical background rather than including them in the hypotheses if it is not going to be tested.

6. The presentation of hypothesis 3 on page 8 mentions "conservative messages against public health messages". However, only reliance on specific information sources was measured in this study. There is no strong argument made to support that certain information sources are always conservative. The authors assume that communication by religious leaders, friends and family to be conservative, which may not always be true. There is no clear presentation of exactly which sources are considered to be conservative vs. liberal, and no strong rationale presented for the assumption either. In the discussion, the authors must also address the critical role of social norms on people's vaccination decisions. It may be almost impossible to change its salience in decision making. In addition to recommending improvements to public vaccination campaigns, the authors should also address how governments can capitalize on the important role on social norms and reduce misinformation among groups by collaborating with religious leaders to communicate accurate information.

7. Page 20, paragraph 2 discusses about the depoliticization of vaccination campaigns in the US. Prior to presenting this argument, it would be important to first clearly present research that establishes the politicization of vaccine campaigns in the US. Without it, it only appears to be an assumption by the authors.

8. Throughout the paper, in discussing hypothesis 3, the authors use messages and messengers (i.e., sources of information) interchangeably. The study only measures reliance/preference of messengers but not the actual content of the messages shared.

9. One significant improvement that would re-position this article's importance would be the examination of interactions between individuals' demographics and their reliance on specific information sources in predicting vaccination behavior. Does the reliance on specific information sources by minorities have differential impact on their vaccination behavior? Are minorities more likely to rely on one or more types of information sources than others? If yes, vaccination campaigns should focus on these sources to deliver accurate and fact-based information to these populations.

Fifth, there are also some concerns related to the presentation of the methods and results sections -

1. A power analysis supporting that the current sample is sufficiently powered for the analyses in this study is missing.

2. On page 10, for most of the measures mentioned, it would be good to include an example item per measure, the format of the response scale (i.e., Likert type scale, etc), response anchors and measures of reliability for any multi-item measures.

3. The analyses presented on page 11 can be included as a separate section titled, "Analyses".

4. Some results from the same analysis are presented as odds ratios and others as percentage. In the spirit of consistency, it would be better to present one or the other and to also indicate in the description clearly what the reference group for each of the analyses is and that the differences are significant.

5. Page 19, paragraph 1 includes a sentence that appears to be inaccurate, "Other interesting findings indicate that individuals who have nurses for the main information source about vaccines are 68% less likely and those who mainly receive information from family and friends are 68% less likely to be fully vaccinated for COVID-19, compared to the CDC category."

6. Page 23, paragraph 2 refers to this as an observational study when in fact it is a correlational study.

Overall, there is a need and scope for re-positioning the purpose of this article significantly. This data would indeed be a valuable addition to the research on vaccination behavior, especially given the WEIRD problem affecting medical research in the US. However, in my opinion, significant changes to the current draft may be required.

6. PLOS authors have the option to publish the peer review history of their article (what does this mean? ). If published, this will include your full peer review and any attached files.

**Do you want your identity to be public for this peer review?** For information about this choice, including consent withdrawal, please see our Privacy Policy .

Reviewer #1: No

Reviewer #2: No

---

## [Author Response · Author response to Decision Letter 1]

2 Jan 2025

Please see the "Response to Reviewers" document for our specific reviewer and editor comments.

---

## [Decision Letter · Decision Letter 1]

12 Feb 2025

PONE-D-24-37455R1A Comprehensive Analysis of COVID-19 Vaccination Behavior: The Influence of Religion, Information Sources, Political Leanings, and Demographic FactorsPLOS ONE

Dear Dr. Over,

Thank you for submitting your manuscript to PLOS ONE. After careful consideration, we feel that it has merit but does not fully meet PLOS ONE’s publication criteria as it currently stands. Therefore, we invite you to submit a revised version of the manuscript that addresses the points raised during the review process.

Thank you for your great efforts in addressing the majority of the concerns raised by the two reviewers in Revision 1. 

Since the second reviewer was unable to review your revision, and PLOS One requires at least reviewers to evaluate the revised manuscript, we have invited a third reviewer to provide their assessment before considering the manuscript for publication. We apologize for any inconvenience this may cause and kindly request you to fully address all concerns raised by the third reviewer.

We look forward to receiving your revised manuscript.

Kind regards,

Weijun Yu, Ph.D., M.D., M.S.

Academic Editor

PLOS ONE

Journal Requirements:

Reviewers' comments:

Reviewer's Responses to Questions

**Comments to the Author**

1. If the authors have adequately addressed your comments raised in a previous round of review and you feel that this manuscript is now acceptable for publication, you may indicate that here to bypass the “Comments to the Author” section, enter your conflict of interest statement in the “Confidential to Editor” section, and submit your "Accept" recommendation.

Reviewer #1: All comments have been addressed

Reviewer #3: (No Response)

2. Is the manuscript technically sound, and do the data support the conclusions?

Reviewer #1: Yes

Reviewer #3: Yes

3. Has the statistical analysis been performed appropriately and rigorously? 

Reviewer #1: Yes

Reviewer #3: Yes

4. Have the authors made all data underlying the findings in their manuscript fully available?

Reviewer #1: Yes

Reviewer #3: Yes

5. Is the manuscript presented in an intelligible fashion and written in standard English?

Reviewer #1: Yes

Reviewer #3: Yes

6. Review Comments to the Author

Reviewer #1: The reivewer's comments have been appropriately addressed, with major revisions/improvements made. No further reviewer comments needed.

Reviewer #3: Thank you for the opportunity to review the manuscript titled 'A Comprehensive Analysis of COVID-19 Vaccination Behavior: The Influence of Religion, Information Sources, Political Leanings, and Demographic Factors'. This is an important and timely study addressing a critical issue with implications for public health communication strategies. The research provides valuable insights into vaccine hesitancy and its determinants, which are crucial for shaping future vaccination campaigns.

Below are some comments for revision:

Authors should consider revising the introduction to create a more cohesive structure, combining the overlapping sections into a single, streamlined narrative that flows logically from background to research questions and hypotheses. In the Methods and Statistical Analysis sections, clearer differentiation is needed. The Methods section should provide more detail on survey design, data collection, and participant demographics, while the Statistical Analysis section should focus on the techniques used, such as regression models. It would be helpful to elaborate on how the sampling frame and demographic quotas were met, and discuss any limitations in the sampling process. While the absence of a traditional response rate is understandable, a brief discussion on its potential impact on generalizability would be beneficial. Additionally, a clearer justification for the use of multinomial logistic regression is needed, particularly explaining the decision to collapse partially vaccinated and unvaccinated individuals into one group, and how this choice may affect the interpretation of results.

Authors should consider clarifying and streamlining the presentation of demographic data, particularly when describing ethnic/racial groups and vaccination rates, to enhance readability and ensure consistency in how categories are introduced. A clearer connection between the unadjusted and multivariate models would also be helpful, especially when discussing the implications of the unadjusted results. Including a brief discussion of sample limitations, such as small sample sizes for certain groups like Asians or "Other" ethnicities, would address potential concerns regarding generalizability.

Further interpretation of variables like religion and political party affiliation could enhance the discussion, helping to explore the underlying reasons for the observed differences in vaccination rates. In terms of statistical clarity, simplifying the odds ratio interpretations would improve comprehension. For instance, instead of stating "67.3% [(0.327-1)*100; p<.01] less likely among White people," authors should consider revising the phrasing of the odds ratio interpretations to use the actual odds ratios rather than the term "likely," which may cause confusion." Finally, clarifying how the dropped categories for Asians due to small sample sizes might affect the results and whether any adjustments were made to address potential bias would strengthen the analysis.

7. PLOS authors have the option to publish the peer review history of their article (what does this mean? ). If published, this will include your full peer review and any attached files.

**Do you want your identity to be public for this peer review?** For information about this choice, including consent withdrawal, please see our Privacy Policy .

Reviewer #1: No

Reviewer #3: No

---

## [Author Response · Author response to Decision Letter 2]

28 Mar 2025

Reviewer #1: The reivewer's comments have been appropriately addressed, with major revisions/improvements made. No further reviewer comments needed.

Thank you for your encouraging review of our manuscript.

Reviewer #3: Thank you for the opportunity to review the manuscript titled 'A Comprehensive Analysis of COVID-19 Vaccination Behavior: The Influence of Religion, Information Sources, Political Leanings, and Demographic Factors'. This is an important and timely study addressing a critical issue with implications for public health communication strategies. The research provides valuable insights into vaccine hesitancy and its determinants, which are crucial for shaping future vaccination campaigns.

We thank the reviewer for their encouraging and helpful comments which significantly improved the presentation of our argument and results.

Below are some comments for revision:

Authors should consider revising the introduction to create a more cohesive structure, combining the overlapping sections into a single, streamlined narrative that flows logically from background to research questions and hypotheses.

After carefully considering the restructuring of these sections to have a more streamlined narrative, we revised both sections. While we kept “Introduction” and “Literature Review” as two separate sections, we made changes to both. Specifically, we shortened the Introduction, moved some information previously included in it to the first and third paragraphs of the Literature Review, and restructured its flow.

In the Methods and Statistical Analysis sections, clearer differentiation is needed. The Methods section should provide more detail on survey design, data collection, and participant demographics, while the Statistical Analysis section should focus on the techniques used, such as regression models.

In accordance with the reviewer’s comments, we subdivided the previous “Data and methods” section into “Methods” and “Statistical analysis” sections to clarify the differentiation.

It would be helpful to elaborate on how the sampling frame and demographic quotas were met, and discuss any limitations in the sampling process. While the absence of a traditional response rate is understandable, a brief discussion on its potential impact on generalizability would be beneficial.

Thank you. We added information about CloudResearch Prime Panels in the new “Methods” section. We also provide information about demographic quotas in this section.

Additionally, a clearer justification for the use of multinomial logistic regression is needed, particularly explaining the decision to collapse partially vaccinated and unvaccinated individuals into one group, and how this choice may affect the interpretation of results.

The models included in our manuscript do not illustrate multinomial logistic regressions. To address the reviewer’s comments, we clarified in the “Statistical analysis” section that we estimated models for different specifications of the dependent variable. However, we included only the logistic models illustrated in the manuscript.

Authors should consider clarifying and streamlining the presentation of demographic data, particularly when describing ethnic/racial groups and vaccination rates, to enhance readability and ensure consistency in how categories are introduced.

The discussion pertaining to the descriptive statistics has been streamlined to highlight the major patterns in the data. Please see changes in the analysis and results sections.

A clearer connection between the unadjusted and multivariate models would also be helpful, especially when discussing the implications of the unadjusted results.

We appreciate this comment and ensured that the main full results are better integrated with the results of the unadjusted models.

Including a brief discussion of sample limitations, such as small sample sizes for certain groups like Asians or "Other" ethnicities, would address potential concerns regarding generalizability.

Thank you. We added information about limitations of small sample sizes for specific race/ethnicity groups and main source of information for COVID-19 vaccination at the end of the “Statistical Analysis” section, related to Table 4 and models on Tables 5 and 6. We also explain this limitation and our approach to interpret the findings in the paragraphs in which we analyze Table 6 in the “Main results” section.

Further interpretation of variables like religion and political party affiliation could enhance the discussion, helping to explore the underlying reasons for the observed differences in vaccination rates.

In the “Main results” section, when we discuss Hypothesis 2, related to religion and political party affiliation, we explain the findings on Tables 2 and 3. We also include this discussion in the “Final considerations” section. We would like to note that in our original submission of this manuscript to the journal, previous reviewers mentioned that we should not attempt to explore underlying reasons for these differences. The argument was that we would need additional variables related to religiosity and political ideology. We made these adjustments in our first resubmission to the journal in response to that first round of review.

In terms of statistical clarity, simplifying the odds ratio interpretations would improve comprehension. For instance, instead of stating "67.3% [(0.327-1)*100; p<.01] less likely among White people," authors should consider revising the phrasing of the odds ratio interpretations to use the actual odds ratios rather than the term "likely," which may cause confusion."

We have updated the language so readers have a clearer understanding of the results.

Finally, clarifying how the dropped categories for Asians due to small sample sizes might affect the results and whether any adjustments were made to address potential bias would strengthen the analysis.

We added information about limitations of small sample sizes for specific race/ethnicity groups and main source of information for COVID-19 vaccination at the end of the "Statistical Analysis” section, related to Table 4 and models on Tables 5 and 6. We also explain this limitation (such as the dropped categories for Asians) and our approach to interpret the findings in the paragraphs in which we analyze Table 6 in the “Main results” section.

---

## [Decision Letter · Decision Letter 2]

16 Apr 2025

A Comprehensive Analysis of COVID-19 Vaccination Behavior: The Influence of Religion, Information Sources, Political Leanings, and Demographic Factors

PONE-D-24-37455R2

Dear Dr. Over,

We’re pleased to inform you that your manuscript has been judged scientifically suitable for publication and will be formally accepted for publication once it meets all outstanding technical requirements.

Kind regards,

Weijun Yu, Ph.D., M.D., M.S.

Academic Editor

PLOS ONE

Additional Editor Comments (optional):

Reviewers' comments:

Reviewer's Responses to Questions

**Comments to the Author**

1. If the authors have adequately addressed your comments raised in a previous round of review and you feel that this manuscript is now acceptable for publication, you may indicate that here to bypass the “Comments to the Author” section, enter your conflict of interest statement in the “Confidential to Editor” section, and submit your "Accept" recommendation.

Reviewer #3: All comments have been addressed

2. Is the manuscript technically sound, and do the data support the conclusions?

Reviewer #3: Yes

3. Has the statistical analysis been performed appropriately and rigorously? 

Reviewer #3: Yes

4. Have the authors made all data underlying the findings in their manuscript fully available?

Reviewer #3: Yes

5. Is the manuscript presented in an intelligible fashion and written in standard English?

Reviewer #3: Yes

6. Review Comments to the Author

Reviewer #3: (No Response)

7. PLOS authors have the option to publish the peer review history of their article (what does this mean? ). If published, this will include your full peer review and any attached files.

**Do you want your identity to be public for this peer review?** For information about this choice, including consent withdrawal, please see our Privacy Policy .

Reviewer #3: No

---

## [Editor Report · Acceptance letter]

PONE-D-24-37455R2

PLOS ONE

Dear Dr. Over,

I'm pleased to inform you that your manuscript has been deemed suitable for publication in PLOS ONE. Congratulations! Your manuscript is now being handed over to our production team.

Kind regards,

on behalf of

Dr. Weijun Yu

Academic Editor

PLOS ONE